# Upregulation of APAF1 and CSF1R in Peripheral Blood Mononuclear Cells of Parkinson’s Disease

**DOI:** 10.3390/ijms24087095

**Published:** 2023-04-12

**Authors:** Kuo-Hsuan Chang, Chia-Hsin Liu, Yi-Ru Wang, Yen-Shi Lo, Chun-Wei Chang, Hsiu-Chuan Wu, Chiung-Mei Chen

**Affiliations:** Department of Neurology, Chang Gung Memorial Hospital Linkou Medical Center and College of Medicine, Chang Gung University, Taoyuan 33302, Taiwan

**Keywords:** oxidative stress, *APAF1*, *CSF1R*

## Abstract

Increased oxidative stress and neuroinflammation play a crucial role in the pathogenesis of Parkinson’s disease (PD). In this study, the expression levels of 52 genes related to oxidative stress and inflammation were measured in peripheral blood mononuclear cells of the discovery cohort including 48 PD patients and 25 healthy controls. Four genes, including *ALDH1A*, *APAF1*, *CR1*, and *CSF1R*, were found to be upregulated in PD patients. The expression patterns of these genes were validated in a second cohort of 101 PD patients and 61 healthy controls. The results confirmed the upregulation of *APAF1* (PD: 0.34 ± 0.18, control: 0.26 ± 0.11, *p* < 0.001) and *CSF1R* (PD: 0.38 ± 0.12, control: 0.33 ± 0.10, *p* = 0.005) in PD patients. The expression level of *APAF1* was correlated with the scores of the Unified Parkinson’s Disease Rating Scale (UPDRS, r = 0.235, *p* = 0.018) and 39-item PD questionnaire (PDQ-39, r = 0.250, *p* = 0.012). The expression level of *CSF1R* was negatively correlated with the scores of the mini-mental status examination (MMSE, r = −0.200, *p* = 0.047) and Montréal Cognitive Assessment (MoCA, r = −0.226, *p* = 0.023). These results highly suggest that oxidative stress biomarkers in peripheral blood may be useful in monitoring the progression of motor disabilities and cognitive decline in PD patients.

## 1. Introduction

Parkinson’s disease (PD) is a progressive neurodegenerative disorder characterized by tremors, slowness of movement, and freezing of gait [1]. The loss of dopaminergic neurons in the substantia nigra of the ventral midbrain accompanied by the presence of eosinophilic cytoplasmic inclusion bodies (Lewy bodies) enriched with α-synuclein is a pathological hallmark of PD [2]. This neurodegeneration is associated with a deficiency of dopamine in the striatum. The exact pathogenesis of PD is unknown, but oxidative stress is thought to play a role in the neurodegeneration associated with the disease [3]. Neurons are particularly vulnerable to oxidative stress due to their high levels of unsaturated lipids, and dopamine metabolism also generates hydrogen peroxide and other reactive oxygen species (ROS) [4]. Iron deposition [5], mitochondrial dysfunction [6], inflammation mediated by microglial activation [7], and reduced levels of antioxidants and antioxidant enzymes [8] also contribute to increased ROS levels, which expose dopaminergic neurons to chronic oxidative stress. Genes linked to inherited PD, such as α-synuclein (*SNCA*), parkin (*PARK2*), DJ-1 (*PARK7*), PTEN induced kinase 1 (*PINK1)*, and leucine-rich repeat kinase 2 (*LRRK2*), are also associated with mechanisms that increase oxidative stress [9]. Identifying oxidative stress or inflammation biomarkers in patients with PD could provide insights into the roles of oxidative stress and inflammation involved in pathogenesis and potentially aid in the detection and monitoring of disease progression.

Cerebrospinal fluid (CSF) is considered to be a good source of biomarkers of PD because it comes into direct contact with the diseased brain tissue. However, CSF is not as easily accessible as other tissues such as blood or peripheral blood mononuclear cells (PBMCs), which can be collected through minimally invasive methods. Several molecules involved in oxidative stress have been identified in PD patients, including elevated plasma levels of homocysteine [10], which are known to cause dopaminergic neuronal loss by inhibiting mitochondrial activity and increasing oxidative stress [11]. High levels of malondialdehyde (MDA), a product of lipid peroxidation, have also been reported in the plasma of PD patients [12,13,14,15]. Increased levels of 8-hydroxydeoxyguanosine (8-OHdG), an oxidized DNA damage marker have been shown in the peripheral blood of PD patients [12,16,17]. A meta-analysis showing elevated concentrations of 8-OHdG, MDA, nitrite, and ferritin, and reduced levels of catalase, uric acid, and glutathione in the peripheral blood of PD patients, further supports increased oxidative stress in PD [18]. The expression levels of nuclear factor erythroid 2-related factor 2 (*NRF2*), an anti-oxidative factor involved in the pathogenic processes of PD [19,20], are also elevated in the PBMCs of PD patients [21]. Activation of the transcription factor nuclear factor kappa B (*NF-κB*) that controls target genes encoding proinflammatory cytokines and chemokines has been shown in brain regions of PD at post-mortem [22]. Microarray analysis of substantia nigra from PD patients also showed that reactive astrocytes appear to be responsible for the activation of microglia which in turn releases proinflammatory cytokines contributing further to neurodegeneration [23]. Serum interleukins (IL-2, IL-10, IL-6, IL-4), tumor necrosis factor-α (TNF-α), interferon-γ (IFN-γ) and soluble TNF-α receptor-1 concentrations were elevated in PD patients [24,25]. Plasma cytokine levels were significantly correlated with the disease severity in PD patients [26]. These studies suggest elevated inflammatory factors in the peripheral blood of PD patients.

Previous gene expression studies have shown that the expression of several oxidative stress- or inflammation-related genes such as heat-shock protein 70-interacting protein (*ST13*), proteasome 20S subunit alpha 2 (*PSMA2*), aldehyde dehydrogenase 1 family member A1 (*ALDH1A1*), BCL11 transcription factor B (*BCL11B*), nuclear-encoded mitochondrial gene (LRPPRC), interleukin 1 beta (*IL-1β*) and complement receptor 1 (*CR1*) was altered in the peripheral blood of PD patients compared to the healthy controls [27,28,29,30,31,32,33]. In this study, we measured the gene expression levels of a panel of genes involved in oxidative stress or inflammation including previously reported differentially expressed genes as described above in PBMCs of PD patients. We further examined if the expression levels of identified genes were significantly correlated with clinical scores of motor or cognitive impairments of PD patients.

## 2. Results

### 2.1. Expression Profiles of Peripheral Blood Mononuclear Cells in the Discovery Cohort for PD

To identify candidate peripheral gene expression markers for oxidative stress for PD, we examined the expression profile of PBMCs using an in-house q-PCR array that included 52 candidate genes involved in oxidative, chaperon, and inflammation pathways in a discovery cohort of 48 PD patients and 25 controls (Table 1). *ALDH1A1* (PD vs. control: 0.063 ± 0.022 vs. 0.053 ± 0.016, *p* = 0.047), apoptotic protease activating factor 1 (*APAF1*, PD vs. control: 0.459 ± 0.188 vs. 0.354 ± 0.082, *p* = 0.006), *CR1* (PD vs. control: 0.073 ± 0.031 μM vs. 0.054 ± 0.032, *p* = 0.026), and colony stimulating factor 1 receptor (*CSF1R*, PD vs. control: 0.375 ± 0.082 μM vs. 0.325 ± 0.094, *p* = 0.032) were significantly upregulated in PBMCs of PD patients (Table 2).

### 2.2. Validation of Candidate Gene Expression Markers in a Validation Cohort

The identified candidate gene expression markers were further validated in an independent validation cohort including 101 PD patients and 61 controls. Scores for Clinical Dementia Rating (CDR), Beck Depression Inventory-II (BDI-II), Hamilton Depression Scale (HAM-D), 39-item PD questionnaire (PDQ-39), Neuropsychiatric Inventory (NPI), UPDRS, and UPDRS part III were significantly higher in PD patients compared to controls (all *p* < 0.001). Conversely, scores for MMSE and MoCA were significantly lower in PD patients (all *p* < 0.001). The plasma level of α-synuclein was significantly higher in PD patients compared to controls (PD: 190.07 ± 159.53 fM, control: 110.70 ± 65.78 fM, *p* < 0.001). The levels of pre-prandial glucose, glycohemoglobin, albumin, triglycerides, and cholesterol were similar between the two groups. When stratifying PD patients by disease severity, we found that those in the advanced stage had a significantly higher mean age than those in the early stage (*p* = 0.003). Scores for CDR, BDI-II, HAM-D, PDQ-39, NPI, UPDRS, and UPDRS part III were also significantly higher in advanced-stage PD patients compared to early-stage PD patients (all *p* < 0.001). Advanced-stage PD patients had lower scores for MMSE, MoCA, and Activities of Daily Living score (ADL) compared to early-stage PD patients (all *p* < 0.001). The detailed results are displayed in Table 3.

In PBMCs, higher expression levels of *APAF1* (PD: 0.34 ± 0.18, control: 0.26 ± 0.11, *p* < 0.001) and *CSF1R* (PD: 0.38 ± 0.12, control: 0.33 ± 0.10, *p* = 0.005) were demonstrated in PD patients compared with controls (Table 2). Furthermore, PD patients at the advanced stage demonstrated higher expression levels of *APAF1* compared with the early stage (advanced stage: 0.43 ± 0.16, control: 0.32 ± 0.18, *p* = 0.011) (Table 2 and Figure 1A). The expression level of *APAF1* was correlated with scores of UPDRS (r = 0.235, *p* = 0.018, Figure 1B) and PDQ-39 (r = 0.250, *p* = 0.012, Figure 1C). The expression level of *CSF1R* was negatively correlated with scores of MMSE (r = −0.200, *p* = 0.047, Figure 1D) and MoCA (r = −0.226, *p* = 0.023, Figure 1E).

## 3. Discussion

This study analyzed gene expression related to oxidative stress or inflammation in patients with PD. Using a two-stage validation process, we found that the expression of *APAF1* and *CSF1R* was increased in PBMCs of PD patients, particularly in advanced-stage patients. Although we have found increased levels of *ALDH1A1* and *CR1* in PD patients compared to the controls from the discovery cohort, these results were not confirmed in the validation cohort. Interestingly, these two genes have been shown to be differentially expressed in peripheral blood in PD by previous gene expression studies [28,29,31]. The inconsistent results may arise from different approaches used. The expression levels of *APAF1* were correlated with scores for parkinsonism, as measured by the UPDRS and PDQ-39, while the expression levels of *CSF1R* were negatively correlated with scores for cognitive function, as measured by the MMSE and MoCA. These findings suggest that peripheral oxidative stress biomarkers may be useful for detecting and monitoring neurodegeneration in PD.

The damages induced by ROS accumulate and trigger the release of cytochrome c from mitochondria during the induction of apoptosis. Cytosolic cytochrome c binds to APAF1, which adopts a heptameric quaternary structure and recruits procaspase-9 to form apoptosome [34]. The overexpression of *APAF1* promotes apoptosis in various cell models [35,36], highlighting its central role in the activation of the intrinsic apoptotic pathway. *APAF1* is highly expressed in the substantia nigra of patients with PD [37]. In accordance with the previous report, our findings showed upregulation of *APAF1* in PBMCs of PD patients. The overexpression of **a** dominant negative variant of *APAF1* suppressed both apoptosis and nigrostriatal degeneration in MPTP-treated mice [38]. The use of **a** pramipexole transdermal patch has been associated with the downregulation of *APAF1* in the 4-phenyl-1,2,3,6-tetrahydropyridine (MPTP)-treated PD mouse model [39]. Apoptosis induced by the overexpression of *LRRK2* variants for familial PD can be prevented by genetic ablation of *APAF1* [40]. Therefore, APAF1 appears to be a potential target for PD treatment through apoptosis inhibition. We further found a correlation between the expression level of *APAF1* in PBMCs and the scores of UPDRS or PDQ-39, suggesting the potential of APAF1 as a biomarker to monitor motor disabilities during PD progression.

Belonging to the class III transmembrane tyrosine kinase receptor family, CSF1R is a major regulator of microglial development and maintenance in the brain [41]. The binding of the ligand, CSF1 or IL-34, induces homodimerization and autophosphorylation of CSF1R to activate phosphoinositide 3-kinase (PI3K)/Akt, protein kinase C (PKC), and extracellular signal-regulated kinase 1/2 (ERK1/2)pathways [42]. The deficiency of CSF1R leads to a significant reduction in microglial density in rodents [43,44]. On the other hand, the overexpression or activation of CSF1R in microglia upregulate the expression of pro-inflammatory cytokines IL-1β, macrophage inflammatory proteins-1α (MIP-1α), inducible nitric oxide synthase (iNOS), and IL-6 [45], which increase the production of ROS [46]. In substantia nigra, *CSF1* and *CSF1R* expression was increased in PD patients compared with controls [47]. CSF1R expression is also upregulated in the striatum of an MPTP-treated PD mouse model [47]. *CSF1* rs1058885 T variant, which is proposed to reduce the CSF1 activity, is less common in PD patients [48]. Treatment with the CSF1R inhibitor GW2580 significantly attenuates microglial activation, dopaminergic neuron loss, and motor behavioral deficits in an MPTP-induced mouse model [47]. Furthermore, our results consistently found the upregulation of *CSF1R* in PBMCs of PD patients. The levels of *CSF1R* were correlated with cognitive impairment measured by MMSE and MoCA. Similar to our results, the upregulation of CSF1R has been reported in the microglia of post-mortem brain samples from patients with Alzheimer’s disease (AD) [49]. Deletion of CSF1R in *APPSwe*/*PS1*AD mice delayed cognitive decline [50]. These studies indicate the involvement of *CSF1R* expression in cognitive decline during neurodegeneration, while targeting CSF1R signaling may be a viable neuroprotective strategy for PD and other neurodegenerative diseases.

This study has several limitations that may affect the results. The sample size may not be large enough to detect small changes in gene expression in PD. The low proportion of patients with advanced PD may also make it difficult to detect differences in gene expression between the two different PD stages. Additionally, the potential interactions of medications taken by the patients may contribute to differences between the groups. However, our study still provides valuable information on gene expression in PBMCs of PD patients and suggests **a** potential therapeutic benefit of inhibiting *APAF1* and *CSF1R* in these patients. Further research on larger and independent patient groups is needed to confirm these findings.

## 4. Materials and Methods

### 4.1. Patient Recruitment

Patients with PD were recruited from the neurology clinics of Chang Gung Memorial Hospital. The diagnosis of PD was based on the United Kingdom PD Society Brain Bank clinical diagnostic criteria by two neurologists specialized in movement disorders (KH Chang and CM Chen) [51]. Controls were recruited from neurology outpatient clinics by a convenience sample of individuals seen at the time of recruitment, and were frequency matched for the sex and age of patients. All subjects received clinical assessment including UPDRS [52], Hoehn and Yahr stage [53], ADL [54], PDQ-39 [55], CDR [56], MMSE [57], MoCA [58], NPI [59], BDI-II [60], and HAM-D [61].

The blood was collected in a PaxgeneTM blood RNA tube (Pre-AnalytiX, Qiagen, Hilden, Germany). Total RNA from leukocytes was extracted using the PaxgeneTM blood RNA Extraction Kit (Pre-AnalytiX, Qiagen) and purified and concentrated using the RNeasy MinElute spin column (RNeasyH MinEluteHCleanup Kit, Qiagen). RNA quality was determined using the A260/A280 absorption ratio and capillary electrophoresis on an Agilent 2100 Bioanalyzer automated analysis system (Agilent, Santa Clara, CA, USA).

### 4.2. Measurement of α-Synuclein in Plasma

We used the immunomagnetic reduction assay to measure the plasma levels of α-synuclein [62]. Frozen human plasma samples were brought to room temperature for 20 min and then mixed with reagents (MF-ASC–0060, MagQu, New Taipei City, Taiwan) for α-synuclein assay. Calibrators (CA-DEX-0060 and CA-DEX-0080, MagQu) and control solutions (CL-ASC-000T, MagQu) were also used in each batch of measurements. The immunomagnetic reduction analyzer (XacPro-S361, MagQu) was utilized to perform duplicate measurements of α-synuclein for each sample. The average concentration of the duplicated measurements was reported, and plasma α-synuclein levels were determined by technicians who were blinded to the clinical diagnosis.

### 4.3. Profiling of Relevant Gene Expression Related to Reactive Oxygen Species (ROS) and Inflammation Using a Quantitative Polymerase Chain Reaction (q-PCR) Array

Reverse transcription (RT) was performed using Superscript III (Invitrogen, Waltham, MA, USA) with an initial concentration of 5 μg total RNA. We established an in-house human panel for ROS profiling analysis using real-time qPCR with SYBR green reagents (Applied Biosystems, Foster City, CA, USA). The thermocycling conditions were as follows: 50 °C for 2 min, 95 °C for 10 min, 95 °C for 15 s, and 60 °C for 1 min for 40 cycles, on the ABI 7900 HT RT-PCR system (Applied Biosystems). Each sample was analyzed in duplicate. Relative expression values were normalized to glyceraldehyde-3-phosphate dehydrogenase (*GAPDH*). Relative gene expressions were calculated using the 2^−ΔΔCt^ method, ΔCt = Ct (target gene) − Ct (*GAPDH*), where Ct indicates the cycle threshold (the fractional cycle number at which the fluorescent signal reaches the detection threshold). The primer sequences for the selected 52 genes are listed in Table 4.

### 4.4. Statistical Analysis

All statistical analyses were conducted using SPSS version 19.0 (SPSS, Chicago, IL, USA). Baseline characteristics and metabolite concentrations are presented as mean ± standard deviation for continuous variables and counts (percentages) for categorical variables. Comparisons between the control group and PD groups, including early PD and advanced PD, were performed using an independent Student’s *t* test or one-way analysis of variance (ANOVA) with Tukey’s post hoc test. The Pearson correlation coefficient (r) was used to analyze correlations between levels of gene expression and age or clinical assessment such as UPDRS, H&Y, ADL, PDQ-39, CDR, MMSE, MoCA, NPI, BDI-II, and HAM-D. A *p* value of <0.05 was considered statistically significant.

## Figures and Tables

**Figure 1 ijms-24-07095-f001:**
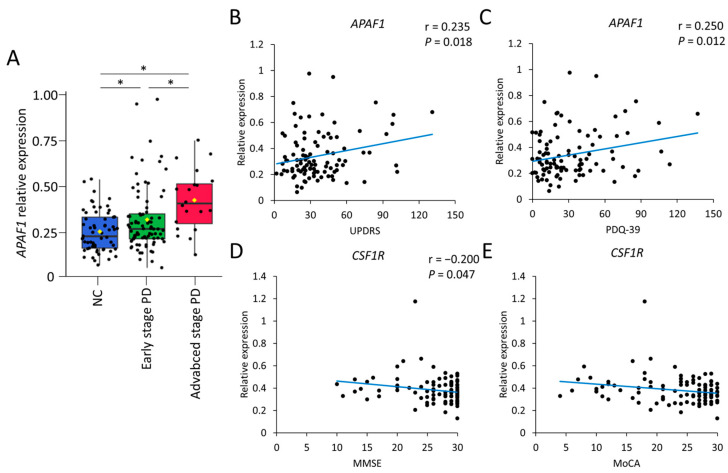
(**A**) Difference in expression level of *APAF1* between Parkinson’s disease (PD) patients at early (N = 81) and advanced stages (N = 20) compared to controls (N = 61). (**B**,**C**) The correlation between expression level of *APAF1* and scores of Unified Parkinson’s Disease Rating Scale (UPDRS) or 39-item PD questionnaire (PDQ-39). (**D**,**E**) The correlation between expression level of *CSF1R* and the scores of mini-mental state examination (MMSE) or Montreal Cognitive Assessment (MoCA). *: Statistically significant between two groups, *p* < 0.05, one-way analysis of variance (ANOVA) with Tukey’s post hoc test.

**Table 1 ijms-24-07095-t001:** Clinical characteristics of the discovery cohort.

	Control (n = 25)	PD (n = 48)
Age (years)	66.33 ± 8.56	68.58 ± 10.79
Male (%)	155 (60.00)	30 (62.50)
Hoehn and Yahr stage		2.14 ± 0.95
LEDD (mg)		849.28 ± 616.80
UPDRS		
Total	1.12 ± 1.88	53.38 ± 31.25 *
Part III	0.16 ± 0.78	32.56 ± 19.60 *

LEDD: levodopa equivalent daily dosage; PD: Parkinson’s disease; UPDRS: Unified Parkinson’s Disease Rating Scale. *: Statistically significant in comparison with control, *p* < 0.05, Two-tailed Student’s *t* test.

**Table 2 ijms-24-07095-t002:** Expression levels of 52 genes in the discovery cohort of Parkinson’s disease patients (PD) and controls.

Gene	Control (N = 25)	PD (N = 48)	*p* Value
*ALDH1A1*	0.053 ± 0.017	0.063 ± 0.023	0.047
*CR1*	0.054 ± 0.033	0.073 ± 0.031	0.026
*CSF1R*	0.325 ± 0.096	0.375 ± 0.084	0.032
*APAF1*	0.354 ± 0.084	0.459 ± 0.190	0.002
*HIP2*	0.043 ± 0.014	0.046 ± 0.012	0.408
*EGF*	0.009 ± 0.010	0.006 ± 0.003	0.153
*HSPB8*	0.003 ± 0.005	0.001 ± 0.001	0.128
*PSMA7*	0.580 ± 0.126	0.564 ± 0.149	0.639
*BCL11B*	0.148 ± 0.067	0.144 ± 0.057	0.818
*ST13*	1.075 ± 0.319	1.053 ± 0.519	0.821
*C3*	0.010 ± 0.003	0.011 ± 0.004	0.117
*CSF1*	0.080 ± 0.038	0.092 ± 0.070	0.349
*APOA1*	0.001 ± 0.000	0.001 ± 0.000	0.808
*FAS*	0.018 ± 0.010	0.021 ± 0.013	0.330
*CR2*	0.007 ± 0.003	0.006 ± 0.004	0.309
*C3AR1*	0.152 ± 0.087	0.144 ± 0.075	0.680
*NLRP1*	0.527 ± 0.137	0.548 ± 0.200	0.601
*NLRP3*	0.074 ± 0.026	0.087 ± 0.038	0.116
*Nrf2*	0.596 ± 0.136	0.650 ± 0.222	0.200
*HMOX1*	0.248 ± 0.082	0.264 ± 0.084	0.435
*HMOX2*	0.112 ± 0.040	0.094 ± 0.031	0.058
*GCLM*	0.016 ± 0.009	0.018 ± 0.013	0.326
*BAD*	0.020 ± 0.007	0.018 ± 0.007	0.323
*NQO1*	0.014 ± 0.016	0.014 ± 0.025	0.888
*LRPPRC*	0.007 ± 0.010	0.006 ± 0.010	0.684
*PSMA2*	0.006 ± 0.004	0.007 ± 0.004	0.414
*PPARGC1A*	0.005 ± 0.010	0.004 ± 0.013	0.771
*C4A*	0.007 ± 0.003	0.007 ± 0.004	0.680
*C4B*	0.103 ± 0.168	0.103 ± 0.181	0.998
*C1qA*	0.026 ± 0.014	0.026 ± 0.016	0.986
*C5AR1*	3.929 ± 1.122	4.022 ± 2.105	0.807
*CASP1*	0.475 ± 0.179	0.570 ± 0.253	0.067
*IL-1beta*	0.347 ± 0.208	0.399 ± 0.258	0.352
*CASP3*	0.092 ± 0.043	0.096 ± 0.038	0.723
*CASP6*	1.178 ± 5.757	0.024 ± 0.013	0.326
*HSF1*	0.167 ± 0.041	0.162 ± 0.055	0.683
*HSP90AA1*	0.274 ± 0.129	0.270 ± 0.118	0.914
*OPA1*	0.022 ± 0.012	0.022 ± 0.010	0.820
*MFN1*	0.037 ± 0.024	0.040 ± 0.018	0.557
*MFN2*	0.069 ± 0.028	0.081 ± 0.033	0.115
*HSPA5*	0.047 ± 0.032	0.057 ± 0.060	0.334
*HSPB1*	0.227 ± 0.098	0.211 ± 0.119	0.542
*HSPD1*	0.229 ± 0.085	0.200 ± 0.085	0.181
*DNM1L*	0.019 ± 0.010	0.020 ± 0.008	0.588
*BCL2*	0.071 ± 0.034	0.069 ± 0.034	0.796
*BID*	0.490 ± 0.303	0.478 ± 0.156	0.849
*GSS*	0.049 ± 0.009	0.047 ± 0.012	0.688
*GSR*	0.180 ± 0.055	0.188 ± 0.053	0.550
*IKBKB*	0.100 ± 0.030	0.088 ± 0.028	0.122
*CASP9*	0.046 ± 0.023	0.045 ± 0.022	0.857
*DNAJB1*	0.250 ± 0.078	0.257 ± 0.087	0.749
*NRF1*	0.011 ± 0.005	0.012 ± 0.004	0.825

**Table 3 ijms-24-07095-t003:** Demographic characteristics and blood biochemical parameters of the validation cohort.

	Control (n = 61)	PD		
		Early Stage (n = 81)	Advanced Stage (n = 20)	Total (n = 101)
Age, year	67.18 ± 8.06	65.31 ± 10.72	73.00 ± 9.18 ^b,c^	66.83 ± 10.87
Male, No (%)	30 (49.18 %)	42 (51.86%)	11 (55.00%)	53 (52.48%)
Age at onset		61.78 ±15.83	61.05 ± 13.78	61.63 ±15.45
Duration of PD		5.47 ± 7.94	12.31 ± 6.23	6.83 ± 8.10
BMI	24.14 ± 3.04	24.51 ± 3.45	23.67 ± 3.39	24.35 ± 3.45
Pre-prandial Glucose, mg/dL	99.32 ± 11.16	105.40 ± 22.67	99.35 ± 19.59	104.20 ± 22.22
Glycohemoglobin, %	5.87 ± 0.42	6.00 ± 0.66	5.81 ± 0.37	5.96 ± 0.62
Albumin, mg/dL	4.43 ± 0.20	4.46 ± 0.22	4.24 ± 0.25	4.42 ± 0.24
Total cholesterol, mg/dL	184.14 ± 28.94	177.05 ± 35.36	166.70 ± 29.19	175.00 ± 34.47
LDL-C, mg/dL	107.84 ± 27.71	104.35 ± 31.09	95.15 ± 30.32	102.52 ± 31.75
HDL-C, mg/dL	55.13 ± 13.99	53.19 ± 15.53	53.50 ± 12.73	53.25 ± 15.02
Triglyceride, mg/dL	103.87 ± 63.02	112.74 ± 60.98	89.35 ± 37.45	108.11 ± 57.86
Creatinine, mg/dL	0.78 ± 0.22	0.89 ± 0.38	0.79 ± 0.24	0.87 ± 0.36
MMSE	29.51 ± 0.88	27.47 ± 3.69 ^b^	21.75 ± 6.50 ^b,c^	26.34 ± 4.95 ^a^
MoCA	27.56 ± 4.26	24.59 ± 5.54 ^b^	17.70 ± 8.33 ^b,c^	23.23 ± 6.77 ^a^
CDR	0.20 ± 0.24	0.31 ± 0.24 ^b^	0.68 ± 0.36 ^b,c^	0.39 ± 0.31 ^a^
BDI-II	1.64 ± 2.86	6.26 ± 1.89 ^b^	15.40 ± 7.64 ^b,c^	8.07 ± 6.63 ^a^
HAM-D	1.61 ± 2.68	5.15 ± 3.68 ^b^	10.35 ± 6.52 ^b,c^	6.18 ± 4.86 ^a^
ADL	98.11 ± 10.79	99.69 ± 1.44	71.00 ± 27.18 ^b.c^	94.01 ± 16.69
PDQ-39	5.72 ± 8.15	23.09 ± 18.22 ^b^	66.95 ± 32.36 ^b,c^	31.77 ± 27.92 ^a^
NPI	0.52 ± 1.68	1.98 ± 2.83	7.20 ± 7.28 ^b,c^	3.01 ± 4.61 ^a^
UPDRS				
Total	1.54 ± 2.17	28.46 ± 13.26	70.45 ± 25.15 ^b,c^	36.68 ± 23.94 ^a^
Part III	0.38 ± 1.20	17.63 ± 9.26	42.42 ± 15.19 ^b,c^	22.44 ± 14.49 ^a^
LEDD		445.91 ± 429.28	1306.55 ± 65.57 ^d^	616.34 ± 592.05
α-Synuclein (Femtomolar)	110.70 ± 65.78	186.22 ± 157.41 ^b^	205.64 ± 166.69 ^b^	190.07 ± 159.53 ^a^
Relative gene expression				
*APAF1*	0.26 ± 0.11	0.32 ± 0.18 ^b^	0.43 ± 0.16 ^b,c^	0.34 ± 0.18 ^a^
*CR1*	0.10 ± 0.06	0.12 ± 0.08	0.10 ± 0.06	0.12 ± 0.07
*CSF1R*	0.33 ± 0.10	0.38 ± 0.13 ^b^	0.37 ± 0.08 ^b^	0.38 ± 0.12 ^a^
*ALDH1A1*	0.08 ± 0.03	0.10 ± 0.09	0.8 ± 0.04	0.10 ± 0.08

ADL: Activities of Daily Living; BDI-II: Beck Depression Inventory II; BMI: body mass index; CDR: Clinical Dementia Rating; HAM-D: Hamilton Depression Rating Scale; HDL-C: high-density lipoprotein-cholesterol; LEDD: Levodopa Equivalent Daily Dose; LDL-C: low-density lipoprotein-cholesterol; MMSE: Mini-Mental State Examination; MoCA: Montreal Cognitive Assessment; NPI: Neuropsychiatric Inventory Questionnaire; PD: Parkinson’s disease; PDQ-39: 39-item PD questionnaire; UPDRS: Unified Parkinson’s Disease Rating Scale. ^a^: Statistically significant in comparison with controls, *p* < 0.05, Two-tailed Student’s t test. ^b^: Statistically significant in comparison with control, *p* < 0.05, one-way analysis of variance (ANOVA) with Tukey’s post hoc test. ^c^: Statistically significant in comparison with PD at early stage, *p* < 0.05, one-way analysis of variance (ANOVA) with Tukey’s post hoc test. ^d^: Statistically significant in comparison with PD at early stage, *p* < 0.05, Two-tailed Student’s *t* test.

**Table 4 ijms-24-07095-t004:** Genes and primers of q-PCR array related to oxidative stress and inflammation.

Gene Symbol		Gene Primer (5’→3’)
*ALDH1A1*	aldehyde dehydrogenase 1 family member A1	F-gttgtcaaaccagcagagca R-caagtcggcatcagctaaca
*CR1*	complement C3b/C4b receptor 1	F-cccgaactctgcaaacaaat R-gtttagcacgaggcagaagg
*CSF1R*	colony-stimulating factor 1 receptor	F-aaggtggctgtgaagatgct R-ccttccttcgcagaaagttg
*APAF1*	apoptotic peptidase activating factor 1	F-tggccttcagcagttctttt R-gggagcaggaatagtgtcca
*HIP2*	ubiquitin-conjugating enzyme E2 K(UBE2K)	F-ggtgtggcacagtttgtcag R-caccacaacaaagcaccatc
*EGF*	epidermal growth factor	F-tagctcagtgcagcctcaaa R-gcaccatggctaatgcctat
*HSPB8*	heat shock protein family B (small) member 8	F-acagccaggaagtcacctgt R-ggcctaacacaaccaagcat
*PSMA7*	proteasome 20S subunit alpha 7	F-tgatggcactcctaggctct R-gcctcatgacagcaagttca
*BCL11B*	BCL11 transcription factor B	F-ggagggaatgggagagaaag R-ggatcatctgcttccgtgtt
*ST13*	ST13 Hsp70 interacting protein	F-ctggaatgcctggactcaat R-aggttgcttttccttcagca
*C3*	complement C3	F-ggaaaaggaggatggaaagc R-caatggccatgatgtactcg
*CSF1*	colony-stimulating factor 1	F-ggagacctcgtgccaaatta R-ggccttgtcatgctcttcat
*APOA1*	apolipoprotein A1	F-tggatgtgctcaaagacagc R-tcacctcctccagatccttg
*FAS*	Fas cell surface death receptor	F-gccacctttcttttctgcaa R-actggagagcagacagcaca
*CR2*	complement C3d receptor 2	F-caaggcacaattccttggtt R-ctccaggtgcctctttcttg
*C3AR1*	complement C3a receptor 1	F-tcccttcctttatgccctct R-gtttttgaagtccgctgctc
*NLRP1*	NLR family pyrin domain containing 1	F-ccagaaacctgaaggagctg R-tgagcacattgaagctcagg
*NLRP3*	NLR family pyrin domain containing 3	F-aaaggaagtggactgcgaga R-ctggtttaccaggccaaaga
*Nrf2*	NFE2 like bZIP transcription factor 2(NFE2L2)	F-catgccctcacctgctactt R-tgttctggtgatgccacact
*HMOX1*	heme oxygenase 1	F-tccgatgggtccttacactc R-taaggaagccagccaagaga
*HMOX2*	heme oxygenase 2	F-tccggtagtccctgtttttg R-ttctgggtgagagggatgag
*GCLM*	glutamate-cysteine ligase modifier subunit	F-tcctacctgcaccctcaact R-tgtgaacatcagcctggaaa
*BAD*	BCL2 associated agonist of cell death	F-caggcctatgcaaaaagagg R-taaacctggctcgcgactta
*NQO1*	NAD(P)H quinone dehydrogenase 1	F-ttactatgggatggggtcca R-tttcaatgcaccacaagagg
*LRPPRC*	leucine-rich pentatricopeptide repeat containing	F-cttggcccagtggacatagt R-gaggctgaggcacaagaatc
*PSMA2*	proteasome 20S subunit alpha 2	F-gccctcttcgctatcagatg R-accaacaggaaccagcaaac
*PPARGC1A*	PPARG coactivator 1 alpha	F-tttccttttgccatggaatc R-gaaagaaccgctgaacaagc
*C4A*	complement C4A (Rodgers blood group)	F-cccaatatgatccctgatgg R-ccactgctctgtcttgtcca
*C4B*	complement C4B (Chido blood group)	F-acggcttccaggttaaggtt R-ctcctcgatccagctattcg
*C1qA*	complement C1q A chain	F-gaaatctgcctgtccatcgt R-gcagatgggaagatgaggaa
*C5AR1*	complement C5a receptor 1	F-atgccatctggttcctcaac R-caggaaggagggtatggtca
*CASP1*	caspase 1	F-tgttcctgtgatgtggagga R-tgcccacagacattcatacag
*IL-1beta*	interleukin 1 beta	F-ttcgacacatgggataacga R-tctttcaacacgcaggacag
*CASP3*	caspase 3	F-tttttcagaggggatcgttg R-cggcctccactggtatttta
*CASP6*	caspase 6	F-gaagcaggttccctgttttg R-ctcccaaagtgctgggatta
*HSF1*	heat shock transcription factor 1	F-gacataaagatccgccagga R-ctgcaccagtgagatcagga
*HSP90AA1*	heat shock protein 90kDa alpha family class A member 1	F-ggcagaggctgataagaacg R-ttcttccatgcgtgatgtgt
*OPA1*	OPA1, mitochondrial dynamin-like GTPase	F-ccacagatttctcccaagga R-attactgtggggcatggaga
*MFN1*	mitofusin 1	F-tggggctgtgagctcttaat R-acactccttggtggttccag
*MFN2*	mitofusin 2	F-tgttggctcagtgcttcatc R-aagtccctccttgtcccagt
*HSPA5*	heat shock protein family A (Hsp70) member 5	F-tttcacagtgcccaagagtg R-tgatcactcactccccatca
*HSPB1*	heat shock protein family B (small) member 1	F-acgagatcaccatcccagtc R-tttgacaggtggttgctttg
*HSPD1*	heat shock protein family D (Hsp60) member 1	F-ttcaggttgtggcagtcaag R-tggtcacaatgacctctcca
*DNM1L*	dynamin 1-like	F-cagtgtgccaaaggcagtaa R-gatgagtctcccggatttca
*BCL2*	B-cell CLL/lymphoma 2	F-aagattgatgggatcgttgc R-gcggaacacttgattctggt
*BID*	BH3 interacting domain death agonist	F-gcaggcctaccctagagaca R-tccatcccatttctggctaa
*GSS*	glutathione synthetase	F-accgctcgtctctttgacat R-ttgccagcttctttggtctt
*GSR*	glutathione reductase	F-agtgggactcacggaagatg R-ttcactgcaacagcaaaacc
*IKBKB*	inhibitor of nuclear factor kappa B kinase subunit beta	F-agcatgaatgcctctcgact R-gccgtgaaactctggtcttg
*CASP9*	caspase 9	F-ttccctcattttgctccaac R-tggtgcacgcctgtagtaag
*DNAJB1*	DnaJ heat shock protein family (Hsp40) member B1	F-acagtgaacgtccccactct R-agtccttggggagctcagat
*NRF1*	nuclear respiratory factor 1	F-gtggcaggacttctttctgc R-taattccatgcgggtttcat
*GAPDH*	gyceraldehyde-3-phosphate dehydrogenase	F-cgagatccctccaaaatcaa R-ttcacacccatgacgaacat

F: forward primers, R: reversed primers.

## Data Availability

The data presented in this study are available on reasonable request from the corresponding author.

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
