# Peer review of "Upregulation of APAF1 and CSF1R in Peripheral Blood Mononuclear Cells of Parkinson’s Disease"

_ijms, 2023, doi:10.3390/ijms24087095_

Round 1

Reviewer 1 Report

Review of manuscript ID: ijms-2323533

 Title: Upregulation of APAF1 and CSF1R in peripheral blood mononuclear cells of Parkinson’s disease, by Chang et al.

General comment: the study shows the clinical assessment of different markers related to redox status, inflammation, apoptosis and proliferation in peripheral blood mononuclear cells of patients at two different stages of Parkinson’s disease. Amongst all biomarkers evaluated, the authors focus on two particular genes, APAF1 (apoptotic peptidase activating factor 1), that is positively correlated with scores for Parkinsonism, and CSF1R (Colony Stimulating Factor 1 Receptor), that is negatively correlated with scores for cognitive function. Since the former is found upregulated in PD patients while the latter is found down-regulated the authors conclude that their results confirm previous findings and support the interest of both genes and their corresponding proteins as targets for PD therapeutics. An attractive clinical assay where hypothesis and objectives are sound, methods seem adequate and results are clearly exposed and fairly discussed. Some minor comments are detailed below:

Specific comments:

1)      Line 40; remove PD after reference 9.

2)      Line 53; MDA should be spelled out as malondialdehyde and its interest as a lipid damage marker (as with 8-OHdG) stated.

3)      Line 85 and tables 1 and 2; Normal Controls sounds redundant, perhaps healthy controls (as stated in introduction by the authors) or simply Controls seems enough.

4)      Table 1; UPDRS should be spelled out in the legend.

5)      Table 3, all liters in left column should be in capital L, dL.

6)      Line 220; it should be Hoehn and Yahr.

Author Response

    We would like to express our gratitude to all the reviewers for providing constructive comments that helped us to enhance the quality of the paper. We have made the necessary corrections to the manuscript based on the reviewers' feedback. Please find attached our response to the reviewers, which addresses their comments point by point. The revised sections have been highlighted in yellow for easy reference.

Sincerely yours,

Chiung-Mei Chen

Reviewer 2 Report

This is an interesting, well-conducted and written manuscript  identifying 2 genes involved in oxidative stress and neuroinflammation in blood cells of PD patients. As a strength, the authors verified their initial finding s 4 changed genes on a larger, independent cohort and then narrowed it down to 2 genes. There are just a few minor changes (including language) required which I list below:

1. line 16 and other places (lines 139/141): the term "positively" is not really required (although often used) for correlation since it is automatically a direct one while in the opposite case (as in line 18) it would be an "inverse correlation". Also, please introduce the abbreviation NC here since this is only explained in the methods part as "normal control" while it can also mean "negative control", thus it is important to clearly describe this when first used.

2. line 19 Please introduce the abbreviation MMSE in the abstract as you did for other clinical parameters in the abstract since you only do this further down (line 149) but you should do it when you first use it..

3. Line 35: rather use the term "dysfunction" instead of "abnormalities"

4. Line 37: What you really mean are increased ROS levels as mechanisms such as higher levels of antioxidants rather de-toxify ROS and so not influence their production/generation. Please amend.

5. line 38/9: When referring to genetic changes and genes that are involved in familiar/inherited forms of PD then clearly separate that from spontaneous/idiopathic PD which is the vast majority of cases.

6. line 73: please change "genes involving ox stress" to "genes  (that are) involved in oxidative stress".

7. line 93: I cannot see any blood biochemical parameters in table 1 while  it also shows clinical PD parameters which are not demographic ones and LDD is a medication/treatment. Please rephrase.

8. line 102 "in A validation..."

9. line 104: remove "the"

10. line 105: Please introduce the abbreviations for: CDR, BDI, Ham-D and NPI which you only introduce in the legend to table 3.

11. lines 106, 108, 111: remove "the" in front of "PD".

12. lines 107,108,115 and 116, remove both "the", lines 113, 114: remove "the"

13. table 3 after alpha-synuclein: What is "fM"?

14. lines 145-148: Please remove all "the" as they are not required but introduce one in line 146 in front of "UPDRS".

15. Please state whether you used an ANOVA test for the comparisons in figure 1A since these are 3 groups and thus an ANOVA is required with an appropriate post-hoc test. Also, the figure legend should be beneath the figure.

16. line 166: Importantly, there is NO accumulation of ROS as they are extremely short-lived and only their resulting damage can accumulate. Please correct and also add "during induction of apoptosis" since this is what you describe.

17. line 172 "of A DN...", line 174 "of A...patch"

18. line 177 "appearS"

19. line 182 "THE.... CSFR1"

20. line 123: Is it possibly rather "A major regulator"?

21. line 184: Please remove "the" in front of "homodimerization..."

22. lines 191 and 195: "of/in A...mouse model"

23. line 192 "THE CSF1..."

24. line 193 "with THE ...inhibitor"

25. 198 "HAS been reported.."

26. lines 203+ I am not even sure that the low number of advanced PD cases is really a limitation of your study since it is always rather desirable to identify early changes which might rather be causally involved in early disease than late changes which might rather happen during disease progression, but it might depend on your actual reserach aim. Just consider this notion.

27. line 208: please remove "the" in front of PBMCs.

28. line 209 better "A potential..."

29. lines 249/50: please do not use capital letters for actin and write it in italic (for genes) while large letters are used for proteins rather. Also, in line 250 use a small "t" in Ct and subscribe it.  

30. 

Author Response

(The authors gave the same response as above.)
